# The Landscapes of Sustainability in the Library and Information Science: Systematic Literature Review

Anna Małgorzata Kamińska [1,*], Łukasz Opaliński [2] and Łukasz Wyciślik [3]

1   Institute of Culture Studies, University of Silesia in Katowice, ul. Uniwersytecka 4, 40-007 Katowice, Poland
2   Scientific Information Center, Rzeszow University of Technology Library, 35-959 Rzeszów, Poland; lopa@prz.edu.pl
3   Department of Applied Informatics, Faculty of Automatic Control, Electronics and Computer Sciences, Silesian University of Technology, 44-100 Gliwice, Poland; lukasz.wycislik@polsl.pl
*   Correspondence: anna.kaminska@us.edu.pl

**Abstract:** In times of real threats to the continuity of the human civilization resulting from environmental degradation, depletion of natural resources, overpopulation, and other adverse factors, the issue of sustainable development is the subject of interest of many scientific disciplines. As a leading objective of this paper, the authors take up the topic of sustainable development seen through the lenses of the library and information science, which is considered with special attention paid to its economic, social, environmental, and cultural dimensions. In addition to reviewing the most important literature, the authors also explore the subject matter from a quantitative perspective. As a result of the research, the authors identify the key areas that affect libraries as cultural and scientific institutions, in which work related to the sustainability concept is actively carried out. Quantitative research allowed to determine the proportions of efforts made by scientists within the previously selected areas, and to outline trends observed within those areas—that is, to identify which areas have recently been gaining importance, and which may have ceased to be exploited. The authors hope that the research results not only shed light on the landscape of world science in the subject matter, but above all, that they support contemporary researches of these fields by identifying potentially the most important works influencing the shape of particular research areas, and the identification of current trends, which are present within the mentioned areas as well. Further research directions, which are potentially worth undertaking, are also emphasized.

**Keywords:** information; library; LIS; literature review; SLR; sustainability; sustainable development; buildings; collections; culture; education

## 1. Introduction

The term "sustainability" itself was used for the very first time in 1953. It appeared in the journal Land Economics, in an article by Joseph L. Fisher [1], and was referred to as to a certain rate, or an amount of some natural resource, which is exploited for some reason, and in spite of this, its availability lasts over a long period of time [2], see also: [1]. However, the original idea of "limits to growth", which underlies the whole concept of sustainability and sustainable development, and which would later take the form of the contemporarily well-known multidimensional sustainability idea, was expressed much earlier by Thomas Malthus in 1798 in his population growth theory [3]. The present and modern understanding of this concept has been though fully and explicitly articulated for the first time in 1987, within the frame of the so-called Brundtland Report on Sustainable Development, which is sometimes termed shortly "Our Common Future".

It is also a widely held belief that this concept is composed of three main components, namely, of an environmental one, an economic, and a social one. This three-fold structure is sometimes called the "triple bottom line", or the "three pillars" or "three axes" arrangement [3–5]. The pillars were ultimately and formally established during the Sustainable

Development Congress in Johannesburg in 2002 [6]. Moreover, sustainability is often considered not only a scientific idea, but a moral value, a normative goal based on this value, and a pathway for international policy or social movements as well [7–9]. In its most widespread perception, sustainability is defined as "a development that meets the needs of the present without compromising the ability of future generations to meet their own needs" [10]. A similar notion was proposed by Asheim in the World Bank Policy Research Working Paper [11], which refers to sustainability as to "a requirement of our generation to manage the resource base such that the average quality of life that we ensure ourselves can potentially be shared by all future generations" [11]. It is possible to ascertain that, in general, the emergence of sustainability ideas has been triggered by the growing global awareness about environmental risk, the discernible need to combat climate change, and the scale of the impact of human activity on the biosphere, as well as the diminishing amount of the available natural resources [8].

Other milestones in consolidating and promoting the concept of sustainability are the United Nations (UN) Conference on Environment and Development, held in Rio de Janeiro in 1992; the Kyoto Protocol, signed in 1997; the UN Conference on Sustainable Development, also held in Rio de Janeiro in 2012, during which a document called "The Future We Want" was signed; and announcing the 17 Goals for Sustainable Development (SDGs) by the UN in 2015, which are to be achieved globally by 2030. The full length of SGDs was given in a UN agenda entitled "Transforming our world: the 2030 Agenda for Sustainable Development (UN 2030 Agenda)". Among the stakeholders who signed the UN 2030 Agenda were representatives of the International Federation of Library Associations and Institutions (IFLA), who took the view that including access to information issues in the agenda is of crucial importance for the success of the 17 SDGs' worldwide implementation [3,7,8,12,13]. It is also notable that the first official statement for a commitment to environmental sustainability in the higher education sector—The Talloires Declaration—which was signed in 1990 during a conference held in Talloires, France, and gave birth to the Association of University Leaders for a Sustainable Future (USLF), has exerted an important impact on The Green Library Movement as well. In fact, this was the decisive factor which forced academic libraries to start going green. However, it was not until 2003 when the movement gained its full popularity in the library profession [14,15]. "Green libraries" are, according to IFLA guidelines, environmentally friendly and are also aimed at drawing the public attention to the environmental dimension of the sustainability concept [13]. The full history of the movement, its origin, and foundations are described in detail by Antonelli [14]. The Talloires Declaration has also pushed libraries to introduce new services, e.g., educational outreach programs intended for the broad public [16,17], see also: [14,18]. The IFLAs' Statement on Libraries and Sustainable Development, issued in 2002, can presumably be seen as a continuation of the diffusion of sustainability ideas beyond their place of origin, and as a mark of the ongoing penetration of the whole LIS subject field by them, at the same time [19]. The same can be said about the 2016 IFLA International Advocacy Programme (IAP), whose aim is to support and instruct librarians how to promote and achieve SDGs within their own nearest, private, or vocational surroundings [7].

All SDGs are further subdivided into more narrowly formulated goals, which embrace several different dimensions related to the three mentioned components of the general sustainability concept. Their list includes, but is not limited to, the following tasks: protecting the natural environment; ending hunger and poverty; improving education; sustaining balanced economic growth; reducing inequalities between regions and societies; ensuring peace, justice, and strong institutions for all the people around the world; introducing responsible consumption and production of goods principles; etc. [3,8,9,20]. Furthermore, the importance of implementing cross- and interdisciplinary research agendas in favor of achieving SDGs is often emphasized, which is why library and information science (LIS), as an example of an interdisciplinary field, ought to be included in the selected scientific efforts aimed at supporting SDGs and accomplishment [9]. This is especially evident regarding

the social component of sustainable development because of the social and cultural nature of libraries as institutes, i.e., agencies, that are meant to provide their communities with opportunities to learn, develop, and increase access to various types of information [5]. What is more, it was also noticed that information is an inherent part of every development and innovation, which is why sustainability issues ought to be a prominent research topic within LIS [20,21].

It is also worthwhile to mention that the subject of sustainability is nowadays among the most often discussed themes in the LIS domain, although it was not always the case. One of the first authors who introduced the sustainability issue to librarians and information scientists was Amanda Spink [22]. She elaborated on the role of information science for a sustainable development in the future, especially regarding its economic dimension, which seems to be a prominent one in the face of the information explosion phenomenon [3,9,22]. Spink concluded that the most significant challenge for LIS is to contribute to the debate about the sustainable future, but also to debate the character of sustainable LIS itself [22].

Apart from Spink's work, there is one more early paper which explicitly includes the concept of sustainability into the field of LIS. Boris Elepov and Olga Lavrik [23] determined and explained the role of information science in entering the path towards sustainable development and the role of libraries, which, in the present context, become fundamental institutions of a social nature. They are, in fact, providers of indispensable knowledge, which is one of the key notions related to the sustainable development of civilization, technology, and culture. Information has to be retrieved, assessed, stored, provided, and used to solve the problems one encounters taking the course towards (especially social) sustainability. There is a wide variety of such problems, and their reach is worldwide, which is why it is of crucial importance to ensure an equal and global access to valuable information resources [23]. Libraries are also regarded as institutions inspiring "the spiritual life of a society", which means that libraries, in general, have a potential for creating public opinion, e.g., an opinion on the necessity to reach the mentioned path of sustainable development [23]. If so, at the end of the path, we should see a rise of a new civilization, where ecological economics is the main binding force for the whole global system [23]. The authors also lay emphasis on investigating the library from the perspective of the theory of culture, which would eventually lead to the emergence of the fourth pillar of the sustainable development concept, namely, a cultural one (see Section 3).

As we can see, the concept of sustainable development of different areas of the LIS has evolved over time, and the continuous civilization development of humanity, especially in the technological dimension, gives many opportunities, but also challenges, to the ever-wider adaption of the achievements of humankind. However, which of the areas of application of the concept of sustainable development are of greatest interest to researchers presently? Which of them are often considered together? Which of them are rising, and which, if any, are starting to fade away? We try to answer these questions in this paper.

## 2. Materials and Methods

To control the quality of the study conducted, the systematic literature review (SLR) framework was chosen. It has been recognized as an efficient and powerful way for evaluating and disseminating evidence for studies of this kind, and is regarded as minimizing bias by adopting a transparent process of review studies that supports reproducibility [24]. To maintain consistency with former recent systematic literature reviews published in the field of sustainability (e.g., [25–27]), the three-stage approach to SLRs proposed by Tranfield et al. [24] was adopted, as shown in Figure 1. The SLR was conducted from July to December 2021.

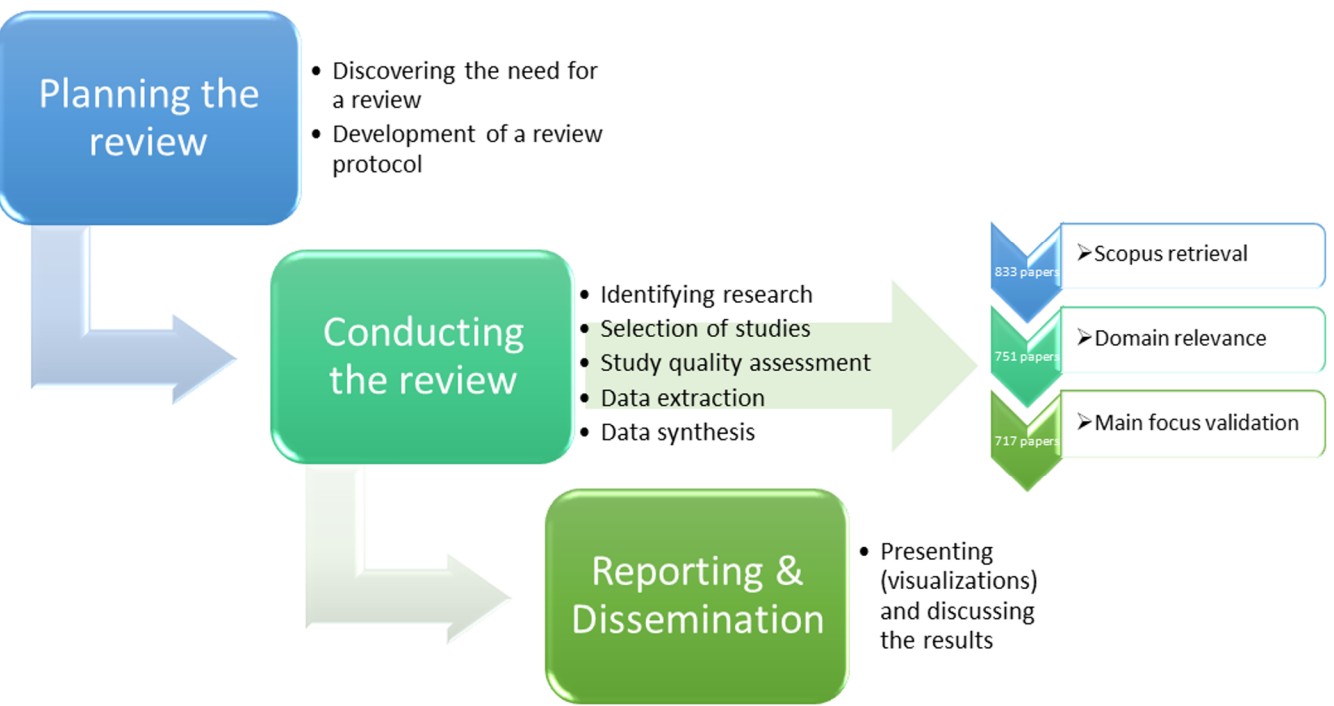

**Figure 1.** Systematic literature review process (adapted from [24]).

### 2.1. Stage 1: Planning the Review

Before starting work implementing the SLR methodology, the need for the research described in this paper was identified and verified. The very idea of the research question formulated in Section 1 was born during the conceptual discussions, whereas its verification occurred through a detailed study of the most important world literature in the field in question. The results of these studies constitute a separate part of this article (Section 3) in which the most valuable works, according to the authors, are pointed out and broken down into topical areas resulting from that descriptive review.

For conducting quantitative research, the Scopus-Elsevier platform was selected, which is believed to be one of the most comprehensive and complete bibliographic data sources, and has been recognized as including more high-quality, peer-reviewed publications than other databases [25]. Based on the interface provided by the Scopus website, a search query was defined to enable the retrieval of bibliographic records:

> ( TITLE-ABS-KEY ( sustainab* )
>     AND ( TITLE ( library OR librari*
> OR "information scienc*" ) )
> )
> AND ( LIMIT-TO ( DOCTYPE , "ar" )
>     OR LIMIT-TO ( DOCTYPE , "cp" ) OR LIMIT-TO ( DOCTYPE , "ch" )
>     OR LIMIT-TO ( DOCTYPE , "bk" )
>     )

As can be seen above, the search query is quite broad—the database is searched for works containing the stem of the word 'sustainability' in the title, abstract, or keywords, and at the same time, in the title, the word 'lis' or the stem of the 'library' or 'information science' words. The search result is also limited to publication types that are articles or conference papers, book chapters, and books. Therefore, the defined search query retrieved 833 records (as of 4 July 2021).

Some concerns could be brought by the form of the search query, namely its certain generality. The point is that, on the one hand, the search query should return a small

number of records that are irrelevant to the entire downloaded corpus (the authors dealt with such cases by manually verifying each of the downloaded records), but on the other hand, that the query should not omit records relevant to the subject matter. There are many considerations on that issue in the literature, from quite sophisticated approaches [28] applied to the cases where the subject matter definition is complex, to the straightforward ones [29] in cases of subject matters with relatively crisp boundaries. The authors decided to follow the latter approach, as a sustainability subject is very recognizable in the area of scientific interest. Despite the existence of many synonyms for the term "sustainable development", the commonness and general knowledge of this generic form allow authors to assume that if it was not included in the keywords section of a given article, it was done on purpose, and not because of carelessness or forgetfulness, which would indicate that the given author is intentionally not positioning their work in the area of sustainable development. In addition, keeping the search query simple results in two other benefits—it facilitates the portability of similar researches to other bibliographic databases (usually having slightly different query grammar), and facilitates the research reproducibility that could be conducted in the future to verify whether the trends of sustainable development in the field of LIS identified in this paper are still valid.

### 2.2. Stage 2: Conducting the Review

The raw dataset obtained from the Scopus database was subjected to further screening by the authors to assess the domain relevance of the scientific works retrieved. At this stage, 82 records were rejected. In the second step, the remaining scientific works were assigned to individual topical areas, in such a way that each of them had to be assigned at most to one area as the main topic covered in its content, and optionally to other areas as side topics. Of the 751 papers for 34, the main area of focus was outside the previously identified 6 topical areas.

Data processed in such a way constituted a database saved to a file in CSV format, so that it was possible to further analyze the data using spreadsheet applications and other analytical tools. Next, the data were analyzed in terms of the frequency of assignment to particular topical areas, the frequency of this assignment to particular areas broken down by the year of publication, and the frequency of entries in the keywords sections.

### 2.3. Stage 3: Reporting and Dissemination

In accordance with the proposal of Tranfield et al. [24], the results and dissemination of the findings were divided into two parts. The first one presents the complete descriptive analysis of the gathered research material. Starting with the characterization of the dataset by metadata description, a summary review is provided by means of tables and charts with supplemental explanations. Then, the second part of the results shows an overview of key emerging topical areas and their trends. That part also relates to the research question that guides this study.

## 3. Descriptive Review of the Literature

At the very beginning of this paragraph, it seems advisable to clarify that the two basic notions, which are the axis around which the rest of the paragraph, as well as the following ones, is arranged, i.e., "sustainability" and "sustainable development", are nearly synonymous. As a consequence, they are oftentimes used interchangeably in the subject literature, and the basic difference between them appears to lie in the broadness of the scope of their meaning. More strictly speaking, "sustainable development" embraces the notion of "sustainability", i.e., it refers to a development of a society, human culture, or their certain aspects, and activities aimed at pursuing the accomplishment of the abovementioned SDGs, with the special emphasis placed on their potential for making all of them "sustainable".

A detailed review of the literature carried out as part of this research shows that the interest in sustainable development research within the LIS domain can be roughly categorized into several main groups or topics. They are in partial accordance with the

eight subjects indicated by Meschede and Henkel [20] on the grounds of their literature review based on a Scopus database [20].

The authors' current proposition of division of the main topics that the sustainable LIS literature is composed of is slightly different from Meschede and Henkel's typology for several reasons. First of all, the authors tried to let the topics emerge, in some way, by themselves during the automated process, and afterwards, the manual literature review and selection. Secondly, Meschede and Henkel's sample of publications was much smaller than the sample used by the authors of the present article, and despite this fact, Meschede and Henkel developed as many as eight categories of topics in contrast to six categories, which came up as a result of the authors' analysis. What this means is that Meschede and Henkel's topics are much narrower and more specialized, which makes it difficult to decide which of them are the most appropriate ones for a particular publication. In other words, the authors' typology is able to grasp more papers of similar content into one category, whereas Meschede and Henkel's scheme is, in the authors' opinion, too constraining in terms of its capability for merging similar topics into one category. For example, Meschede and Henkel differentiated between information and communication technology systems, and information science in general [20]. In comparison to the authors' proposition, it seems that such diversification is an obstacle when an article discusses a wider range of issues which are strongly connected to each other, as is oftentimes the case. Besides that, the various problems discussed within the sustainable LIS literature are often overlapping, and that makes the less detailed classification scheme more suitable for the literature review purpose. What is more, the sample size, which was used by Meschede and Henkel, amounted to only 81 papers, which puts the possibility of generalizing them to the whole LIS field into question. One more problem, that arises from the high level of specificity of Meschede and Henkel's scheme, is the very low number of papers which were assigned to some categories, e.g., the "research institutes and universities" category contains only four articles [20]. This fact may indicate that such a category is underrepresented in a sample, or maybe even unnecessary in the context of the pool of documents that tackle constantly evolving and differentiating subject themes. Briefly speaking, the authors' own categorization appears to better suit the adopted methodology, large sample size, and the possibility of extrapolation of their analysis' outcomes to the vast majority of the whole population of sustainable LIS publications. Apart from that, the typology applied in the present article is also more coherent, and thanks to this, it gives a more comprehensive picture of the examined body of literature.

### 3.1. Buildings

The first topical area is concerned with greening library buildings, their environmental friendliness, energy efficiency, and general sustainability principles, according to which, such buildings should be designed or renovated, and throughout which, they are able to act and serve their users in an environmentally responsible way. The concern, as it relates to green library designs, visibly intensified in 2007, when a seminar "Going Green" was held in Chicago, Illinois. The city planners, architects, and librarians attended to share the cutting-edge solutions that were possible to adopt at that time [5,13,14,30]. One of the biggest and most acute challenges in this area is the so-called carbon footprint of libraries, which is connected to their print and electronic material usage, external and internal architectural design, as well as to energy consumption. These issues pertain to using electricity for printing, photocopying, supplying power for information systems and mobile devices, lighting, heating and air conditioning, enhancing the acoustic quality of an interior library's space; but also to consuming water, using water-conserving apparatuses, choice of construction materials, production of material waste or pollution, high maintenance costs, and lack of universal framework for estimating greenhouse gases (GHG) emission [5,31,32]. The green character of library buildings is also influenced by their nearest vicinity, the landscape surrounding them, their proximity to public transportation, and the presence of cycle routes and pedestrian ones, etc. In summary, all of these factors are taken into

consideration by the Green Building Council, which grants the so-called Leadership in Energy and Environmental Design certificates (LEED certificates). The certificates are part of the worldwide green building certification program [5,14–16,30,32]. LEED certificates have been granted since 2000, and examples of libraries that hold this certificate are given by Stoss [33] and Fedorowicz-Kruszewska [7]. It is also emphasized in the subject literature that libraries of this type, i.e., certified ones, are carrying messages to the public, and they are doing this by displaying the best architectural practices in the area of sustainable design, but also in the sphere of social and cultural sustainability [32]. Some authors remark further that libraries still could do more to promote the environmental dimension of sustainable development, e.g., they could provide (borrow) equipment and tools that are meant to help users in controlling their electricity consumption level (i.e., energy meters), or even garden tools, lawn mowers, plant seeds, musical instruments, or bicycles [14,19]. All of this is still situated within libraries' potential for promoting and acting in favor of exigent sustainability issues in the real world.

A much more comprehensive overview of the ways through which public (but also other) libraries could "go green" is given by Miller [34]. Miller proposes and elaborates on a simple, inexpensive, yet effective means of making public libraries environmentally friendly, and in doing so, becoming pronounced examples for their communities. These include, but are not limited to, focusing on obtaining a LEED certificate, and issues pertaining to lighting, cleaning agents, garbage disposal, electricity saving methods, etc. The author also emphasizes the implications of providing green programming to increase the librarians' and public's environmental literacy skills.

Greening library buildings and services can be seen as a part of a wider endeavor, which can be briefly depicted as combating climate changes. A global fight for a sustainable future certainly has to incorporate this issue, and the LIS specialist community is aware of this fact. Some of them indicated and suggested a vast range of operations which could be undertaken to contribute to this enterprise. For example, Charney and Hauke [35] summarized and exemplified such possibilities, and presented selected strategies adopted by several academic libraries. In their opinion, these examples could be treated as "role models", and could be copied as imitable instances.

However, the problem of climate change seems to be too weighty to be solved by any kind of libraries' actions or agendas themselves. It is particularly obvious in comparison with the actions of the whole industrial sector, as well as many other ecological movements. LIS can be only a minor agent in this respect, and this standpoint is also discernible within the range of sustainable LIS literature, as it appears. This is why the authors of the present article focused more on other aspects of the sustainability concept which are affordable for the tools and initiatives at LIS specialists' disposal.

### 3.2. Information

The second main topic which is widely considered as a part of sustainable LIS issues is the one that focuses on access to information, information society, information and communication systems, and information science in general. The definition of an "information society", which takes into account the notion of sustainability, was extensively discussed by Fuchs [36]. He also proposed his own model to explain the meaning of a "participatory, cooperative, sustainable information society" (PCSIS), in which the conception of cooperation as a social process constitutes the foundation for long-term sustainable development. As a principal feature of this cooperation, the author named human-centered technology development, common socio-economic equity, active efforts to support ecological preservation, and, on top of that, prevailing political freedom, and cultural wisdom [36].

Next, Nolin [3] has made a distinction between sustainable information and information for sustainable development, and has indicated that sustainable information fits in the larger project of sustainable development. He defines the latter one, i.e., information for sustainable development, as "clean information" which is implemented in "clean communication technologies" and, thanks to it, contributes to efficient energy usage and

energy saving. This also means that, among others, it is necessary to migrate information resources to new media, which will allow them to survive for future generations. The first one, i.e., sustainable information, is stipulated as follows: (i) it is a resource for political projects connected to any aspect of sustainable development; (ii) it constitutes a contribution to the communicative aspects of sustainable development and integration them into all "walks of life"; (iii) it supports any global-range transformation towards sustainable goals through facilitating effective processing, storage, and sharing (distribution) of information resources. This elucidation also contains the notion of information as a constitutive and driving force in modern society that is capable of making a shift to the development of a more environmentally conscious citizenry [3,21]. What is more, Nolin emphasizes that the Brundtland Report itself makes frequent references to the notion of information, namely, to the idea of unlimited access to information as one of the basic conditions for social equality, to information gathering and sharing procedures, and environmental information, which should be stored in publicly open digital databases. As stated in Agenda 21, information is also a central linkage point between the integration of the three abovementioned pillars of sustainability, as well as the participation of society, government bodies, policy makers, and entrepreneurs in the common efforts in favor of accomplishing sustainable development goals. It has also been said that information science is bound to seek for a framework which could be used to tie together research on the three fundamental facets of sustainability, i.e., an economic, social, and environmental [3,21,37].

The "clean information" conception which was mentioned by Nolin has been further elaborated on, e.g., by Chowdhury [12,21,38] in relation to "green" information retrieval systems (IR) and services, and "green" information technology (IT). Chowdhury's work deals mainly with greenhouse gases (GHG) emissions and the carbon footprint of information systems hardware, and the information services generated, while supporting the whole higher education sector in general. It can be roughly divided into the administrative and operational information involved in the day-to-day academic administration, and scholarly information required during research and strictly scientific activities. He further asks how information systems can be implemented in an economically and environmentally sustainable way, i.e., how to reduce the GHG emission which is generated by the printed materials (books and journals) industry, and by modern digital information resources, and web-based learning and teaching services [12]. A green IS should be designed in such a way that the emission of GHG throughout its whole lifecycle (from content creation to distribution, access, use, and disposal) is significantly minimized [12]. It would be also desirable to adopt a "green" user behavior as well, to attain carbon neutrality in the future [38]. Green user behavior can be also equated, to a certain degree, with sustainable information practices which rely on using appropriate technologies, standards, methods, and policies, so that sustainability could be achieved during the lifecycle of data and information [37]. Chowdhury proposes several methods of greening IR and IT systems, among which, the most promising seem to be: (i) using innovative, improved materials during the manufacturing of IT components; (ii) putting network interfaces to sleep mode at idle times; (iii) making optimal use of hardware and software by consolidating servers through using virtualization software, and reducing waste associated with obsolete equipment. However, the most prospective way seems to be established by a cloud computing model, which can be regarded as an internet-based utility service that allow users to create, manage, share, and store information without the necessity of investing in an expensive and complex infrastructure [12,38]. Cloud computing can thus contribute to the reduction of energy consumption, and to the reduction of financial costs because of the shared use of computing and network resources, diminishment of the amount of pollution produced by the computer industry and its users, and replacement of printed information sources that generate GHG with digital content that is more environmentally friendly, and, for example, dividing stored files into those that are more and less frequently used. Thanks to such a distinction, the less often used content may be moved to less powerful servers that are also less energy consuming. Besides that, cloud computing can optimize the use of computer

servers by using full power only when required, or by enabling users to use what is known as a "thin client", i.e., a computing facility with only minimum processing capabilities, which reduces the socket energy consumption of the end-user devices.

The advantages of cloud data storage and computing are confirmed also by Prince [39]. The author uses an example of the Ex Libris Alma system, whose architecture is cloud-based, and allows for a shift from operations carried on libraries' own hardware to external hosting services. The main asset of this solution is said to be the fact that any concerns over the purchase and maintenance of local server hardware, network administration, and the possibility of mounting this library-designed system on in-house hardware are rendered unnecessary [39]. Another crucial benefit of such a resolution is the elimination of the perpetual software update cycle necessity, as well as scalability, i.e., reducing or expanding system capability at need, easiness of use, and relocating support services to the side of the vendors [39].

Additional opportunities, which emerge from the discussed model, are standardization, in terms of content creation, organization, and processing, which would make the integration of large data sets possible; and, as a second opportunity, a reuse of digital content, as well as IR tools [12,38]. Researchers would then not need to spend computing and energy resources on something which has already been achieved or constructed, e.g., some software applications, various data analytics, raw data sources, etc. In view of Chowdhury, these issues call for further research, and they also need to be promoted, since they constitute the core of the Green IR research agenda [38]. Chowdhury places a particular emphasis on the integration of the three aspects of sustainability, which, in his opinion, are inevitable to put sustainable development into practice, and make it a reality [21]. The target for the economic sustainability of digital IR is to ensure cheaper and easier access to information; the target for social sustainability is to provide equitable access to information to make a well informed and healthy society possible; whereas the aim of environmental sustainability is to minimize the negative influence of GHG on the environment, and suppress climate changes [21]. The author presents a model showing the different facets of sustainability of digital information services in detail and, moreover, a model for undertaking systematic research towards the interrelation of various research topics that are present within this domain. These are, for instance, new business models, new digital copyright exchange hubs, new funding models, novel databases design or methods of measuring the impact of information, emerging legal issues, or information literacy problems [21]. All of these research directions and challenges are interlinked, similarly to the social, economic, and environmental dimensions of sustainability. In general, we need to be aware that any new development for achieving sustainability in one field may have a positive or negative effect on our efforts within another area of sustainability.

Other authors draw their attention to the environmental impact of IR and IT systems in the context of organizations, industry companies, and businesses, and their organizational culture. For example, Jenkin et al. [40] stress that it is essential to understand how Green IR and IS relate to sustainability in organizations, and propose a framework to advance the Green IR and IS research towards this direction. The authors identified numerous factors which influence an organization's environmental strategies, and enriched their list by adding some other motivational forces. They also broadened the research perspective by including environmental management, environmental psychology, learning, and social marketing and social responsibility programs in it. It is also worth noticing that, at least a part of them, directly aim at increasing the environmental orientation of an organization's employers and employees themselves [40]. In short, the framework presented by Jenkin et al. is certainly a multileveled one, and it focuses on identifying actual research gaps, as well as proposing a set of potential guidelines for future scientific endeavors that are to be conducted within this particular area. Among other newly emerging themes, which have gained popularity within sustainability research in LIS domain, there are also the concepts of IS and IR systems evaluation methods. The idea of measuring the level of progress towards sustainability is one of the outcomes of the United Nations recommendations

issued in 1996 under the title "Indicators of Sustainable Development: Framework and Methodologies" [16]. There are, for example, methods that pay special attention to the economic, social, and environmental dimension, but that divide these dimensions further into more complex and more detailed subdimensions (see, e.g., [41]). Sustainability drivers and assessment criteria within the range of such sub-dimensions encompass such values as health and safety hazards, noise, employees' quality and security of work, efficiency and responsiveness to customer needs and market changes, scale of emissions of chemical substances and other pollutants, fuel consumption, waste reduction, percent of product recycled, and the use of bio-degradable materials [41]. The evaluation and measurement of some aspects of sustainability can be also discussed while taking into consideration quality management in the performance of IR or IS services that are commonly used by libraries.

Similarly, Ochôa and Pinto [42] created a LIS Sustainability Assessment Framework, which is meant to evaluate management structures by addressing the social, environmental, economic, and cultural dimensions that exert their impact on a library. The essence of the framework is to further evaluate how these issues (if any) are integrated into the library's overall strategy. This theoretical model also takes into account incentives such as certain risk indicators, stakeholders' interests, holistic perspective of the designed strategy, forms of developing new competencies, the level of understanding of current societal trends, or communicating results of the assessment graphically using a so-called "Value Map" [42]. It is the authors' belief that the process of developing, implementing, and modifying certain indicators of sustainability will be eventually able to create common standards for libraries, and their product or service life cycles [42]. The degree of completeness of such a framework can be naturally broadened by including some other indicators in it. The sustainability assessment framework proposed, e.g., by Jankowska and Marcum [16], encompasses some other factors that are related not only to IS and IR systems evaluation, but also to other elements of the library's operations subjected to the greening strategy. The additional indicators should also provide data on the construction features of library buildings; energy, paper, ink, and other materials' consumption rate; equipment and paper recycling rates; or maybe even on the whole current operational model that is applied in a particular library [16].

One more important aspect of modern sustainability issues which is raised within the information research area is the question about the sustainability of digital libraries (DL). Chowdhury [43] tackles the problem of economic, social, and environmental sustainability of DLs, and specifies the desired targets for each aspect, e.g., (i) cheap and easy access to high-quality digital information in an economic perspective, (ii) equity of access to information in a social perspective, and (iii) reduction of the negative impact that DLs exert on the environment [43]. The author also proposes a sustainable business model for DLs' managers, recommends an open access policy in connection with academic repositories and DLs, and indicates the challenges that are met by DLs' design and functioning. However, above all, Chowdhury demonstrates a generic model, being at the same time a research framework for the sustainable digital libraries of the future. It is basically a model that combines all three pillars of sustainability with (i) user, (ii) data and their content, and (iii) technical infrastructure categories. Specific goals and research issues have been assigned to these categories, which are meant to help us better understand the different aspects of DLs' sustainability, and the numerous interactions between the mentioned aspects [43]. The model includes elements such as stakeholders of institutions which maintain a DL, and available technologies and skills or culture of the public, i.e., so-called "end users" of a DL system. In essence, appropriate sustainable business models supporting DLs should also support the three elementary sustainability pillars, and such, a comprehensive and holistic approach is necessary if we want to maintain a stable system, which would be deprived of contradictory ideas, actions, or strategies. In other words, the model's aim is to prevent a situation, in which, for example, the methods of attaining economic sustainability are in conflict with some social sustainability rules, etc. [4,43].

Another distinctive example of the LIS sustainability theme, which has been studied scarcely, is the deepening of the level of understanding of the motives, demeanors, and habits underlying our common information practices, based on an ethnographic approach. Nathan [44] studied two ecovillages, i.e., communities striving to ground their daily activities in core values strictly correlated with sustainability [44]. The author conducted semi-structured interviews with selected community members, and observed that most of them were unable to adapt their information behavior to better match accepted concepts of sustainability. Among the most challenging factors which caused that inability were technical issues, i.e., access to the internet, electricity supply, internet service providers, an occasional information overload with the addition of the lack of an effective way of filtering out unnecessary information, as well as the physical properties of the information tools that were used by members in connection with their potential for recycling or reuse [44]. The author concluded that information practices, once established, are extremely resistant to change. Developing new, more sustainable practices is certainly difficult even to conceptualize for those who do not have skills to create innovative practices, which calls into question our ability to explicitly, deliberately, and consciously influence our own information practices, especially regarding sustainable information practices, which seem to stand apart from many other daily practices [44]. As a consequence, the role of information professionals is to develop and inform a knowledge base, people's information literacy, and technical infrastructure, which are required to support truly sustainable information practices that would meet present and future needs [44].

*3.3. Collections*

The third significant group of topics that are dealt with within the subject discussed here consists of sustainable collection creation, storage, and management topics. The term "sustainable collection" has been defined by William Walters as a collection "(...) that can be maintained without significant degradation over time (...) with a budget that provides for continued access to serial resources (...) as well as the timely acquisition of important monographic materials (...). In general terms, an economically sustainable collection is the one for which the rate of increase in prices is no greater than the rate of increase in the library acquisitions budget" [45,46]. In other words, sustainable collection development relies on the effort that has to be put into maintaining and securing documents for future generations, as well as into making them at least partly independent from commercial publishers [16].

One of the more general managerial topics related to library sustainability is the preservation of library materials, i.e., the protection of cultural heritage and property through preventing their physical deterioration, damage, and loss of information content [4]. This leading need encompasses activities such as proper housing, treatment, reformatting, or the replacement of stored materials, and these actions should be done with the sustainability ideas in mind. The same is valid for other typical library practices that accompany preservation, such as production of waste (e.g., scrap) materials, finding ways to recycle materials instead of relegating them to a landfill, choosing "greener" options during supply purchasing processes, etc. Jones also addressed the need for an appropriate preservation environment, especially regarding digital preservation methods. Generally speaking, the main concern here is maintaining the balance between power consumption and the need for proper physical and chemical conditions that prevail in selected library interiors [4]. In conclusion, the author states that it is crucial that all aspects of a library system have to be considered when implementing sustainable practices, and this is not limited to the preservation practices themselves [4]. A more comprehensive list of green library operations and recommended practices is given, e.g., in a paper by Kurbanoğlu and Boustany [17].

There is also a matter of pursuing ways that libraries could "green" the adapted practices of collection development. For example, Connell [47] describes three such ways: (i) selection of materials whose content informs green practices, (ii) de-selection with an emphasis on reusing and recycling materials, and (iii) selection of those materials that

conform to the principle of reducing the carbon footprint an institution generates as a whole [17,47,48]. However, there is also a question about the choice between downsizing print collections in favor of electronic ones, as pointed out by Chadwell [48]. According to Connell, printed books are more environmentally friendly than electronic resources [47]; however, this opinion is not commonly accepted (see, e.g., [1]). Chadwell also indicates the discrepancy between engineering sciences and humanities, where faculty is resistant to discard older print volumes or move them to off-site storage. As a result, funds, which could be earmarked for digitalization or purchasing other library materials, are tied to shelving rarely used holdings [48]. Within this range of daily library practice, Chadwell [48], as well as Mitchell and Lorbeer [46] point to financial issues and budget allocation problems. They argue that the so-called "Big Deals", which are formal arrangements concluded between libraries and publishers or the owners of full-text databases that offer access to highly valued scientific content, are not necessarily the best purchasing option. In other words, the economic sustainability of "Big Deals" is called into question. First of all, a group of peer institutions purchase the same content, which is not cost-effective, and could be transformed, e.g., into a system of interlibrary loans. Secondly, there is also a concern that the databases contain irrelevant, or in some other way unnecessary, materials that will be used infrequently. Thirdly, many institutions use spending money from their shrinking budgets on constantly getting more expensive journals, monographs, and other kinds of resources from the world's leading publishers. It has also been estimated that research libraries usually devote from 70 to 80 percent of their yearly acquisition budgets to journals, which means that some academic libraries can afford only a small portion of new and possibly important books published year by year. This is sometimes seen as a diminishing of the economic sustainability of a library collection as a whole [45,46,48]. The economic, social, and environmental sustainability of "Big Deals" is thus dependent on the factors such as: (i) choosing the most relevant and financially affordable publisher's packages; (ii) taking into consideration the time of duration of access to previously purchased resources; (iii) the level of users' satisfaction and gratification that keeps them engaged in their information seeking behavior; (iv) decisions as to the environmental friendliness of storage of printed versus electronic library materials, especially in terms of GHG emissions and the manner of disposing of the outdated stock; and (v) ways of promoting the library's resources and making them widely available for the general public.

Mitchell and Lorbeer [46] wonder how a library's collection could be made relevant and sustainable at the same time. They recommend, in this respect, making use of journal usage statistics, a proactive information gathering that aims at identifying the most needed "core" materials, and this should be done in association with faculty and students. This is partly because of the so-called bundling practice of publishers, who sell multiple work packages under the terms of a license agreement. As Williams put it: "(...) libraries are willing to pay for articles they don't need to gain access to those they do need" [45]. The lowering of interlibrary loan average cost is also a measure which could be undertaken or at least taken into consideration [46]. Generally speaking, a dialogue between a library and their intended public seems inevitable to identify the most urgent information needs, and to try to allocate funds to those resources that are actually the most desired ones. The users thus should be asked for feedback, and faculties need to be encouraged to collaborate with their library "liaisons" [46]. Walters [45], with respect to the same question that was raised by Mitchell and Lorbeer, argues that a collection which is comprised primarily of books instead of journals is far more sustainable, at least in economic terms. The author indicates in this context, that the inflation rate for books is much lower than that for journals [45]. Walters is aware that although this advice can seem reasonable for undergraduate colleges, elementary school, or some public libraries, regarding academic libraries, the situation is very different. The reliance on serials in scientific libraries is evident, and a large scale cancellation of journal subscriptions is not feasible [45]. On the other hand, the author cites studies which showed not only that undergraduates in the sciences tend to prefer books rather than journal articles, but also that faculty members in the sciences are prone

to use books as often as their peers in the humanities [45]. In terms of journals subscribed by large academic libraries, Walters notices that the most crucial research outcomes are concentrated in a relatively small number of key journals, which can be roughly assessed on the basis of citation analysis [45]. It also seems essential to pay attention to the sustainability of access to individual digital journal-related resources. For those, the most important facet of sustainability of access is the perpetuity of access to the purchased digital content, because it is often a temporary lease agreement which provides access only during the subscription years. In other words, the sustainability of access depends on the sustainability of payments [45].

Brodie [31], in turn, speaks in favor of balancing the growth of the collection in relation to physical space and energy consumption as well [31]. The author gives an example of the library of Macquarie University in Sydney, Australia, which is an institution that seeks to become a leader and a "shining instance" of sustainability [31]. Among other distinctive features and activities involved in the novel library management program, the author mentions the introduction of an automated storage and retrieval system (ASRS), which is equipped with a "virtual bookshelf" that allows users to scan the titles wherever they may be held. The adapted policy visibly contributed to the reduction of floor space that would be required in cases of traditional open-shelf access, and to the reduction of the library's GHG emissions significantly. Moreover, the authorities established some agreed collection storage principles which guide decisions regarding materials that are to be stored on the open-shelves, and which should be relegated to the ASRS. Then, collection profiles for each discipline taught at the University were generated. The assumption here was that these data can be useful when informing a library's consultations with academic staff, and when making an attempt to fine-tune the actual location of physical items (holdings) [31]. According to Brodie, the library building and its managerial policies can be treated as a "living laboratory" for further research and learning about how to achieve the appropriate level of sustainability [31]. The trend towards integrating all aspects of library activity has been acknowledged earlier by Jankowska and Marcum [16]. The authors stated that there is a lack of a comprehensive, holistic framework addressing print and digital collections' sustainability level, social and environmental responsibilities of networking services and practices, and the degree to which library buildings comply with established standards, at the same time [16]. An alternative tool, in this respect, is a checklist developed by Klaus Ulrich Werner in 2013, which is meant to be a sort of reference when designing a new library building, or renewing an already existing one, with the greening practice and SDGs implementation in mind. Such a holistic approach seems to be reasonable if one takes into consideration the fact that a sustainable library has to care for all three dimensions of sustainability, and its strategy cannot be too narrowly focused [7]. Some even say that taking all these aspects into account, a library and its whole operation model constitutes one of the "seven plus wonders of sustainability" [14].

A more constricted glance at the sustainability of library collections, but also at other library-related factors, was presented by Beutelspacher and Meschede [19]. Their main concern is environmental sustainability and the means through which a library can contribute to promoting knowledge on environmental protection and environmentally sustainable development. The most powerful mean, in view of the authors' analysis, seem to be a library offer itself, provided that the offer would be focused on the sustainability subject, and would also be presented in a unique and more sophisticated way than usual, e.g., through encouraging shelves arrangement, exhibitions, special events, social media, etc. The authors argue that it would certainly raise general awareness on sustainability and its subtopics, as well as the sensitization of the public ([19], see also: [16]). The sheer need for materials concerning sustainability issues that exist within any library's collection, including technical libraries, is the result of the vast interdisciplinarity of the sustainability concept (see, e.g., [49]). For example, Applin [50] presents a comprehensive bibliography, i.e., a list of fundamental reference books, journal articles, monographs, and websites for libraries, which tend to support widely understood sustainability efforts among their

users, staff, and the broader community (see: [50]). The vast amount, wide thematic range, and common availability of such literature may result in an effective dissemination of information about sustainability principles among the general public.

Another good example of the research devoted to the structure of library collection is given also by Goodchild and Zhao [51], who attempt to measure the level of completeness and relevance of the engineering sciences collection, which is held by McGill University in Montreal. They placed an emphasis specifically on the part of the collection, which deals with a sustainability theme in engineering sciences. Their findings suggest that if collection managers would follow advice which can be drawn from the study, the library could become the central meeting point for interdisciplinary research, and projects related to sustainability would flourish [51].

*3.4. Education*

According to Nolin, one of the most essential ways of moving towards sustainable society is education. This is also the fourth group of topics of crucial importance that are noticeable within the subject literature. The main point of enquiry here is the conviction that the ideals and imperatives of sustainability have to be ingrained in the minds of students, and it is clear that information professionals are also involved in this process [3]. Libraries, in turn, are in an outstanding position to educate and influence the public [4]. Sustainable development should be therefore translated into an imperative of social and ethical issues, and ought to become a part of university curricula. In particular, teaching students sustainable information could take the shape of conveying to them a set of essential ethical values [3].

Chowdhury and Koya [37] refer to education in the context of an iSchools organization (consortium) which was founded in 2005, and intends to be a global hub for education and research. They aim to connect institutions, businesses, and society members using information and communication technologies, as well as to teach them how to collect, manage, interpret, or decode data from various sources which are necessary for achieving SDGs [37]. The idea of educating society itself is directly related to one of the SDGs, according to which, it is of utmost importance to enhance regional and international cooperation on access to science, technology, and innovation, and enhance knowledge sharing through improved coordination among existing mechanisms [37]. In the light of these assumptions, the authors propose four key areas of teaching and conducting research from the perspective of information science and iSchools. These are namely: (i) sustainable information systems and infrastructure; (ii) sustainable information practices; (iii) sustainable information policies and governance; and (iv) sustainable user education, training, and literacy [37]. In summary, the main goal of a sustainable university or any other educational institution is to change the culture and mindset of its students, lecturers, and staff. The role of iSchools, as seen by the authors, is to create environmental literacy programs so that every student and staff member in a university could become environmentally literate. Secondly, the role is to promote and develop the culture by undertaking research on sustainable information management in every scientific discipline and business. Through this measure, iSchools can contribute to the creation of a culture of shared creation, use, access, and understanding of data for sustainable development, especially in terms of the means and possibilities of achieving SDGs. The graduates will then be able to make proper management, research, or professional contributions at their future workplaces in every branch of business or industry towards accomplishing the abovementioned SDGs [37].

Education towards sustainability can be also regarded as a form of "green" information literacy (IL), which is closely related to environmental literacy, and has a tangible impact on the environment. The first one is more of a tool which needs to be used when one is aiming at improving the second one in his or her theoretical knowledge or real-life actions. Sustainable information literacy can thus be tentatively defined as a set of conventional, educational, and technical abilities, which is at the same time expanded to include

sustainable thinking, i.e., being aware of the fact that our information behavior, choices, and actions all have their own ecological footprint which affect the natural environment in a negative way. Thus, individuals have to be more aware of the issue, as well as be ready to act responsively regarding environmental issues [17]. The necessary requirement for preparing appropriate information literacy instruction is a new approach towards making green information literacy widespread. The authors' proposition is two-fold. The first component is to include green operations and practices in the instruction sessions, whereas the second relies on embedding sustainable thinking and attitudes into each possible aspect of IL curricula. It is also about making internet users conscious of the means of going green in accessing, searching, using, selecting, storing, and sharing information. Increasing the awareness of the users should arouse their motivations to act responsibly while using the obtained IL skills, e.g., at work [17]. The teaching techniques, especially those that pertain to sustainable thinking within the frame of IL themselves, should also be taught as an integral part of general IL instruction. According to the authors, embedding sustainable topics, sustainable thinking, and sustainable scholar resources into IL courses would also make them much richer and more interesting not only for students, but for academic staff as well. Such kinds of conducting research ought to be also promoted to gain regular interest from the general public, which will help increase overall awareness of sustainable development issues [17] (see also: [52]).

In the perspective of the conclusions mentioned above, it is possible to say that information literacy in general, but especially the green IL, can be seen as a key to the economic, social, and cultural development of communities, institutions, and even nations. Repanovici et al. [53] showed through their scientometric study of the literature, and a qualitative research study, i.e., a survey, that the nearest efforts towards transforming society into a sustainably developing one should focus on educators and building their own understanding of the whole concept. The authors also demonstrated that IL influences and reinforces the ability to think in sustainability categories, and brings forth sustainable behaviors among the public. Finally, they suggest launching a module within the IL courses that would be devoted to informing students about the connection between the way they use their information and research skills, and the generation of carbon footprint by the information technology equipment [53]. Repanovici et al. also claim that the sustainable IL is situated at the intersection of three core elements of a higher education system, namely research, curricula, and library. This fact, in the authors' opinion, can be regarded as a new possible research direction to be pursued in more detail in the sustainable LIS field [53]. According to Turner [52], the best place to lay emphasis on, within the range of LIS curricula, is the library management course. The sustainability concepts have their counterparts or corresponding areas in the managerial theory that fall into LIS domain. Namely, an economic pillar of sustainability has its analogue in teaching students fiscal management skills and criteria for making purchasing decisions (budgeting). The environmental pillar is reflected in the library space planning courses, as well as in the materials control matter, which is a part of the broader library buildings planning theory. The social pillar is then included within studies of more marketing-oriented aspects of LIS management, such as leadership, quality improvement, visibility of the key sustainability message directed towards the public (e.g., means of promotion and popularization of certain ideas), and the recognition of motivational incentives that are capable of catching one's attention and stimulating his or her behavior [52] (see also: [19]). Whereas it is possible to incorporate the sustainability concept into undergraduate curricula, because students are more prone to adopt the sought behavioral changes before they enter a graduate program, there is also an alternative of integrating sustainability into masters-level courses or even into all the courses required to accomplish a degree. This, in turn, will presumably bring profit in the future, in the form of enabling the LIS graduates to have a deeper understanding of the many sustainability-related impacts that emerge on a daily basis. Apart from this, in Turner's view, reviewing and updating LIS curricula would better position students for developing their own sustainability-informed practices after graduation [52].

Strong [2] generally argues along the same line, although she focuses more on elementary and middle school programs, where teacher-librarians are part of the educational process, and a library performs the role of a rich resource for supporting sustainability education. This conviction is also asserted by the American Library Association, which issued a resolution regarding the quality of library programs in 2012 [2]. If sustainably oriented information literacy begins to be taught at an early stage of children's education, it will have a significant impact on later students' achievements, and will contribute to nation-wide sustainable educational practices and politics [2]. Furthermore, the author proposes some innovative changes that could be implemented into the current model of education. For example, Strong recommends taking up cooperation between school counsellors, media specialists, and librarians; between special educators and librarians; and between school librarians and building (civil engineering) specialists. Generally, the author advises the cooperation and partnership of professional educators grounded in different disciplines or educational sectors, and—in the aftermath of this—the model of cross-disciplinary sustainability education, encompassing the majority of school personnel, including librarians. At the same time, such a model should be fully integrated into the daily life of a school [2].

Jankowska et al. [18] also advocate a similar approach towards teaching sustainability in the higher education sector, with an emphasis placed on academic libraries and LIS schools, and their fulfilment of the role of an active partner in designing a sustainability curriculum. Within the frame of their study, the authors investigated the level of libraries' engagement in sustainability teaching and curricular activities, and named those which are actually put into practice by university authorities [18]. Among the most meaningful ones, there are, e.g., regular information literacy classes, collaboration with other units on campus, building collections devoted to sustainability topics, greening libraries and their surroundings, arranging exhibitions for marketing purposes or cooperation with academic units regarding developing courses, and organizing speeches [18]. The authors also point the focus of sustainability teaching through actual information about literacy curricula content. They included in their list themes such as social diversity, intellectual freedom and the right to free access to information, organizational ethics, open access policy issues, principles of sustainable collection development, and basic principles of greening library buildings, as well as greening information technology or green users' behavior [18]. Jankowska et al. [18] concluded that: (i) there is a growing interest among scholars in sustainability-related themes discussed within the LIS domain; (ii) open access policy with the entitlement to retain author rights seems to be a favorable road to sustainable models of access to information; (iii) there is a noticeable gap between being actively engaged in sustainability movement at an university campus and an absence of specific documents, such as official statements, commitments, or action plans, which would be targeted precisely at sustainability goals, and included in academic libraries' strategic plans at the same time [18]. Nevertheless, the overall picture of the position and involvement of academic libraries and LIS schools in scholarly sustainability practices is in fact encouraging, especially having in mind that the overwhelming majority of surveyed academic libraries employees and LIS school staff expressed a willingness and readiness to support sustainability research, teaching, and its presence in LIS curricula in their professional (vocational) activities [18].

Technology-based teaching and learning techniques nowadays have become increasingly popular, especially amongst youth and adolescents. There is a number of newly emerging technologies, which admittedly could be introduced into IL, IR, or any other kind of class, but one should firstly consider if paying for it is in fact an evidence-based and well-informed decision. Hayman and Smith [54] address this specific problem, and claim that there is often a lack of available up-to-date research on the pedagogical value of such new technologies [54]. In the light of this, they present an evidence-based model for appropriately selecting newly surfaced educational technology and implementing them into practice, especially regarding a potential library instruction. Apart from the expected advantages of emerging technology-based lessons, such technology can also contribute

to making libraries more efficient in terms of space management and staffing financial rationalization, which is a connection to the sustainability concept in its economic and social dimension. Hayman and Smith [54] display their model in the context of an affordance of a technology in question, and desirable learning process outcomes, which are mainly of pedagogical nature. The first step leading towards a well-informed decision is to conduct an evidence search for comparable cases and their actual results. It is through so-called "environmental scanning" that this first step is done. Scanning is a form of human information behavior, which is in essence acquiring, critically evaluating, and, finally, properly applying information to achieve some particular goal. It is a long-known technique which was applied in organizations' decision-making processes, or in the creation of strategic future-oriented blueprints by them [54] (see also: [55]). After this two-fold step is done, it is necessary to continuously support the alignment of pedagogical aims to technological affordances, and seek the best possible evidence that would further confirm that the supposed alignment actually exists. The evidence needed could be obtained from the subject literature (e.g., case studies), from a library's own study, or through the already mentioned environmental scanning. If a library or an educational institution is going to stay abreast of the modern world of technological changes, and, what is more, become a part of a still growing sustainability social movement, library practitioners have to be information-savvy employees who are prepared to foster sustainable decision-making [54]. Hayman and Smith's proposition refers, as it was already mentioned, to the social and economic sustainability of libraries' practices, chiefly through their underlining of a requirement for conforming to, and applying cutting-edge technologies in the library environment, which contributes to the educational effectiveness of staff, as well as to students' efforts aimed at elevating the awareness of significance of the impact that humans exert on the environment. The environmental dimension is also present within this approach, and can be encapsulated as an eventuality of reduction of the negative results of using obsolete equipment, devices, or appliances, which are usually energy-consuming and troublesome in respect to their reliability, working speed, and general effectiveness.

The need for updating and renewing LIS educational programs is also signalized by Goodsett and Koziura [56], who interviewed LIS students regarding their perceived value as a potential library employee in the perspective of knowledge that has been conveyed to them during the time of their studies. The selected group of information professionals was also surveyed and questioned about the possible ways to improve the current level and range of LIS education to better meet the skills of a contemporary, highly trained librarian [56]. The authors ask their research question in the context of the requirements that the job market demands from graduates. According to the authors, in 2012, Forbes named the LIS degree the worst to earn that year [56]. One of the most urgent drawbacks of many LIS curricula, which emerged from the analysis of authors' survey results, is the lack of hands-on experience, or, in other words, the lack of on-the-job training, which calls for more practical, directly job-related courses. Such preparation could take the form of participation in practicums or internships which could be done both in person and online, as well as a form of taking advantage of potential mentoring opportunities with practicing librarians. The next most commonly indicated flaw is the inability to learn how to work with cutting-edge IT technologies, as well as the absence of management and administrative courses [56]. Some participants even claimed that LIS schools ought to be structured around obtaining practical, professional experience instead of learning theory. They argue that experience is an essential prerequisite for success in looking for employment after graduation. The theoretical knowledge should be a sort of necessary background for a prevailing workplace practical instruction [56].

All of the noted flaws of LIS education curricula are of importance for the social aspect of sustainable development. Librarians who are not well prepared to fulfill their responsibilities not only set a bad example for their potential audience, but are also lacking consciousness of the necessity of spreading sustainability ideas among the public. Their incompetency can potentially result in lowering people's knowledge or cognizance of

environmentally hazardous behaviors, whetting their airiness and thoughtlessness when making use of any sort of resources or equipment, regardless of its hypothetical consequences. Generally, education which is deprived of a sustainability factor will inevitably take its effect in the future, and will inhibit progress towards a sustainable society. Moreover, the cultural pillar of sustainable development of a society is also worth mentioning here. Neglecting this facet could possibly be caused by overlooking the meaning of a given society's heritage, roots of its origin, and future prospects, which are indirect consequences of preserving and diffusing this heritage.

*3.5. Culture*

Two years before Elepov and Lavrik [23] mentioned that there is a vital connection between sustainable LIS, or maybe the sustainability idea in general, and a phenomenon of culture, the World Commission on Culture and Development issued a report titled "Our Creative Diversity" [57], which raised the same issue, i.e., a linkage between culture and sustainable development. At a later time, the notion of culture was increasingly often discussed as an aspect of social sustainability or even as an independent field of study. This is the fifth group of the most important themes that belong to the contemporary sustainable LIS area of research. In 2001, Jon Hawkes explicitly introduced cultural sustainability as the fourth pillar of sustainability concept, and paid special attention to the role that culture plays in local planning [58] (see also: [6,59]).

Since that time, other authors have turned their sights to cultural sustainability, and this cultural turn has resulted in many publications on this topic. This new turn appreciates the meaning of language and discourse in sustaining cultural heritage and the world's languages, and cultural diversity. It has been acknowledged that culture constitutes a part of social sustainability, and covers such aspects as social justice and equity, social participation, economic needs and work, and, above all, awareness of the range of sustainability ideas. Nevertheless, at present, it is widely agreed that culture is of equal significance to social, as well as economic and environmental sustainability [59]. Cultural heritage is understood as a source of identity, a local sense of place, and as a certain amount of cultural capital, which has been inherited from previous generations, and can be passed to the generations which are about to come after us. Cultural capital includes tangible (e.g., historical treasures), as well as intangible (e.g., knowledge) human race achievements, which can of course be interlinked [6]. This heritage has to be available for the public in a sustainable way to disseminate it as wide as possible. This purpose is based on the notion of social inclusion, and this aspect entails a drive to globalization, civilization and technology expansion, and increased social mobility. The key problem in this view is to prevent damage that could potentially be done to cultural identity or continuity during its migration among generations. Besides that, not every item can be preserved, and, clearly, the preservation process requires some kind of a selection, which is a question of economic nature [6].

Soini and Birkeland [6] gave a comprehensive summary of a cultural sustainability-related literature, and differentiated seven story lines, each associated with a particular point of view that is taken in a formulation of a cultural sustainability definition and its main characteristics or goals. The authors differentiated the following standpoints regarding this subject: cultural heritage research, cultural vitality studies, economic viability standpoint, cultural diversity matters, locality, eco-cultural resilience proposition, and an eco-culture civilization conception. Eco-cultural trends, e.g., concentrating on nature as a part of civilization, and seeking an integration of human- and nature-made systems [6]. According to the authors, it mostly shows that the fourth pillar of sustainability is now well grounded in the sustainable LIS domain, and it is situated in three main cognitive contexts, namely in a human livelihood (mainly tourism and farming) one, a technological one, and a heritage management methods one. The concept is also seen as a metal-level structure that is able to work at a supra-disciplinary level, which, in a sense, moves beyond scientific disciplines, and reaches a shift in humans' thinking, and the perception of culture and sustainability [6]. A central aim of the cultural turn seems to be to maintain the continuity of culture, and

its accumulation, storage, and preservation. It does not necessarily exclude the theoretical construction of the culture as a reproductive resource to be used in local and regional culture promotion, development, and hypothetical reconstruction, if needed. In this view, cultural sustainability is also considered a support for sustainable economic development. Cultural locality and diversity storylines stress the need for the inclusion of a variety of different perceptions and values with respect for the individual cultural rights of any cultural group, which is a sort of communitarian political orientation. Summarizing, the authors concluded that the cultural pillar of sustainability is an inevitable foundation for meeting the holistic goals of the sustainable development movement, and plays several different and substantial roles in this ongoing process [6]. The roles may vary from poststructuralist deliberations about language being a representation of a culture or its part, and the meaning of the term "sustainability", to knowledge organization systems (KOS), which would be a modern solution to technical aspects of cultural heritage preservation, especially in terms of knowledge and language.

Fraisse et al. [60] express their conviction that language is one of the most important barriers that pose an obstacle to the sustaining of global knowledge diversity. The authors indicate that a number of efforts that have been taken by the LIS community, e.g., KOS' development, making progress in the IRs discipline, outworking of metadata exchanging standards, and multilingual electronic libraries and archival projects, all help to create efficient, cheap, and reliable resources for the purpose of preserving rare or expiring languages [60]. Apart from that, they claim that the language barrier is also a central issue, which has to be addressed by International Society for Knowledge Organization (ISKO) authorities. Along this line of reasoning, Fraisse et al. argue that with the increasingly important globalization context at the background, multilingualism should become one of the major concerns within the boundaries of the LIS field. This is the reason the group of authors decided to present a new paradigm, which is designed for the community of volunteers, who are willing to take part in the knowledge organization process. Being more precise, the idea is about engaging as many widespread contributors as possible, who would add their unique parts of knowledge about translations of some noteworthy original works written in underrepresented languages [60]. The authors actually conducted an experiment in this area, and their choice fell to the famous novel "Adventures of Huckleberry Finn". Over 30 translations were obtained, and on the base of their content, 10 parallel corpora were built. The authors reported further that during the process, 22 under-represented languages were identified amongst all gathered full-text translations [60], which is a sign of the impressive scale of response from end-users, and of a promising potential for the future of this project.

Digital resources and their management are other aspects of the fourth sustainability pillar that are of interest within culture-oriented sustainable branch of LIS. Eschenfelder et al. [61] gave a detailed proposition of a nine-dimensional framework for operationalization of the concept of organizational sustainability for digital resources management. The broader context here is the digital cultural heritage and a community that has worked to establish, sustain, and promote it. The term "organizational sustainability" denotes, in the present context, staff and work practices (organizational practices) that maintain digital content and services availability and long-lasting survivability, given ongoing challenges [61]. The proposed framework stems from the authors' extensive literature review, as well as a review of similar frameworks that already exist. Their nine dimensions encompasses the following factors that determine an organization's sustainability: (i) technology (hardware software, data formats, metadata); (ii) management (market research, strategic planning, business models, stakeholders engagement); (iii) relationships (cost reduction, efficiency, resources sharing, partnerships); (iv) revenue (sources of revenue, fees, grant funding, reputation); (v) costs (expenditures, cost modelling, accountability); (vi) valued product and service (feedback from users, metadata quality, weeding, and acquisition policies); (vii) disaster planning (threats to the continued sustainment of digital files, a hypothetic cease of an institution existence, etc.); (viii) legal policy (facilitating access, capital investment

encouraging, copyrights); (ix) metrics and assessment (evaluation of projects, indicators of usage, business plans, demonstrating impact, risk management) [61]. Identification of these dimensions, along with their operationalization, should lead, as the authors hope, to an improvement in the structural, administrative, or managerial arrangements of an organization (e.g., a library that holds a digital collection), and, as a consequence, to facilitating an evidence-based approach to the digitalized cultural heritage long-term sustainability [61].

One more occurrence of a cultural point of view in the sustainable LIS domain is the role independent libraries play in sustaining human culture in the whole range of its manifestations. Independent libraries, with their origins dating back to the 18th century, were founded as private, or were based on a model that relied on members' fees or financial support from wealthy benefactors [59]. Loach and Rowley [59] ask about an understanding of the contribution that independent libraries make in favor of external cultural sustainability agendas, as well as about the role that an organizational culture plays in achieving sustainability in independent libraries. The authors also aim at contextualizing the notion of cultural sustainability with respect to the wider GLAM (i.e., galleries, libraries, archives, and museums) sector. As a result, the authors developed four core categories that constitute the main points of cultural sustainability relevant to independent libraries. These are: preservation of heritage, cultural identity, cultural vitality, and cultural diversity [59]. The authors also found that the form of organization culture that prevails among independent libraries can be perceived as exclusive, as well as continuity- and tradition-valuing, and not necessarily holding innovation and inclusivity in high esteem. In the authors' view, it is the reason why such libraries should become more commercially minded, and more aware of the external market beyond their traditional user base [59]. Although it may seem that the culture of exclusivity may foster the task of heritage preservation, on the other hand, it jeopardizes the external focus oriented towards a broad audience's cultural vitality and diversity. These conflicting priorities pose a primary challenge to independent libraries' organizational culture.

### 3.6. Others

The sixth, last, and, at the same time, the least consistent group of contemporary sustainable LIS literature is composed of several different, but yet sometimes reoccurring, thematic areas. For example, Audunson et al. [62] examine the role of public libraries as an infrastructure for a sustainable public sphere, which is said to be a necessary precondition of any democratic system. Libraries are, in this context, providers of knowledge, agents facilitating informed citizenry, supporters of a robust public sphere, and, more generally, arenas for public debate [62]. The authors' team determined four key issues that have to be dealt with in any research devoted to the public sphere, and public libraries relations research. The first one is the problem of social inclusion and equity of access to information; the second is promoting democracy, community values, and social capital. The third strand of research is concerned with censorship and freedom of speech, and the fourth relates to social media in a library key message being transmitted in an expanding digital world. These are all mainly political and social factors, but there is also a practical aspect of the research, especially as to the libraries' function in urban and community development, which can be described as a potential moving force. Libraries, as the authors claim, have the potential for recognizing and signaling the quality of a city, and are able to attract people's attention, and, what is more, to expand the potential for encouraging and promoting citizens' democratic participation [62].

Other authors draw their attention to the meaning of librarians' involvement in cross-disciplinary research, i.e., an integration of professionals from many different disciplines into a team, which is treated as a valuable means to achieve some particular SDGs. In Igbinovia's [63] opinion, the phenomenon of cross-disciplinary research is required to overcome the problems of sustainable development, which appear at an interdisciplinary level. Diverse teams of collaborators tend to be more innovative, more creative, and more prone to maximize breakthroughs while minimizing failures at the same time [63]. How-

ever, such involvement, although desirable, is, at present, deprived of research grants, and research findings that were drawn in multi-, inter-, or transdisciplinary collaboration are not implemented into libraries' daily operation schemes. This is especially evident in developing countries, where the level of investment in research is particularly low. In the meantime, a collaborative and common effort of a wide variety of specialists can significantly improve our understanding of SDGs, their future actualization, and a multifaceted approach to them, as well as the problems that occur on the path towards sustainability. In summary, Igbinovia recommends that the regulatory bodies of LIS in Nigeria should encourage and aim at intensifying librarians' engagement in cross-disciplinary research, that the government should consider providing research grants, that appropriate measurement techniques should be applied to capture research implementation outcomes, and that Nigerian librarians should aim at enhancing their information and communication technology skills [63]. Taking these measures is anticipated to contribute to the further global advancement of the sustainable development agenda.

Developing countries are also concerned with other steps that could be taken towards achieving SDGs. For example, Omona [64] remarks that LIS centers in Uganda act as agencies which enhance social and economic development by the creation of an information society, and empowering people to exercise their rights, learn new skills (e.g., information literacy skills), be economically active, and enrich their overall cultural identity. Among the values that can be spread through information services in the marginalized part of a nation, there are also notions of democracy, innovation, economic growth, business success, etc. [64]. Omona concludes that the most important component of Uganda's strategy for sustainable development is to enhance peoples' literacy and, following on from that, the common enlightenment of the general public. A different aspect of the developing countries' situation is the problem of ravaging warfare, which creates unfavorable conditions for any development at all, including sustainable development as well. Olajide [65] discussed the ways by which LIS can serve as a nonviolent and peaceful method of managing conflicts within the borders of developing nations and countries. He indicates that due to a lack of reliable and relevant information, e.g., on electoral processes or the terms of some agreements, a lot of unnecessary conflicts would break out, and gives an example of such conflict that took place in Nigeria [65]. The author stresses that during an attempt of peaceful conflict resolution, lawyers and judges need comprehensive legal information which may prevent the hypothetical miscarriage of justice, or conflict expansion. Olajide also made some recommendations to libraries' and information centers' authorities to enhance their efforts for nation-wide supporting and managing conflict resolutions, and the reconciliation of feuding parties, i.e., equipping libraries with the most current and relevant materials, especially materials on warfare-related topics; organization of seminars and workshops on the same subject; creating conditions for free access to information, etc. [65].

Presumably, there is a need to add one more explanation here. The results presented in the proposed work are part of a much more holistic research in the field in question. A more detailed breakdown of the "Other" category into subcategories would be difficult to implement during the first tagging run the authors did so far, as without a detailed quantitative picture of the given candidate subcategory (namely the frequency of its occurrences), it is inadvisable to include such a division. Instead, in this article, the authors tried to describe a general picture of the category in question without including the definitive results of its quantitative analysis. More particularly speaking, the "Other" category is very heterogeneous, and there is a lot of various topics included in it, which are represented by a diversified number of works. To put it another way, some of its subcategories are represented by merely a couple of papers, whereas others are slightly more popular. For example, several authors dedicated their articles to the most favorable business models, marketing strategies, technological innovations, or personnel's professional competences, which could be helpful when developing a holistic scheme of vocational practices aiming at SDGs accomplishment through some areas of library practice. Other themes tackled

by authors of the papers assigned here took, in turn, a generic look at the whole range of possibilities of making a library a sustainable agency. As can be seen, the works which were grouped into this particular category are usually of a complex thematic range, and they are difficult to assign to some specific, previously distinguished main category of the authors' interest. In fact, this was one of the authors' initial assumptions: that all papers which do not fit into any of the foremost five thematic scopes will be qualified as "Others". This is also the reason why the authors treated this bulk of papers as secondary ones, and planned to devote a separate analysis to them at a later time. To sum up, the connection of the "Other" category to the remaining ones is by no means unambiguous, but rather complicated and problematical. That is to say, there is no one, clear, and explicit link which would allow to recategorize "Others", and split this cluster of papers into a few consistent subgroups. Such a division would have to be done on a much lower level of aggregation of empirical data than that which was adopted in the present article.

To provide a brief summary of the foregoing review, it can be stated that there is a growing number of sustainable LIS literature reviews that can be observed in recent years. This seems to signal that the interest on the sustainability issue within the LIS specialist community is burgeoning, as well as the amount of literature on this topic. There are many different approaches to this theme, and the vast majority of them cover one or more problem areas directly connected to some of the abovementioned pillars of sustainability and sustainable development. However, despite this fact, there is still undoubtedly much more space for subsequent creation, development, and implementation of new sustainable LIS research frameworks, models, problem identifications, or problem solutions, as well as for their further refinement.

## 4. Results and Discussion

In the previous section (Section 3), the matters related to the individual topical areas covered by the issues of sustainable development in the LIS were discussed using the most representative examples found in the scientific literature. This chapter presents the results of quantitative research.

The downloaded and screened dataset, as described in Section 2, was further processed so that its individual records representing scientific works are assigned to individual topical areas in such a way that the record can be assigned to, at most, one topical area (or it may not have any assigned) as the main one, and to many (or none at all) side topical areas. The metadata describing the dataset includes the following data fields:

- Title;
- Year of publication;
- Keywords;
- URL link to the full description of the given work on Scopus website;
- Main topical area;
- Side topical area(s).

The first analysis was made on the basis of the main topical areas discussed in individual scientific works. They are intended to show the proportions of scientific interest between individual topical areas, and are presented in Figure 2.

As can be seen, from a global perspective (i.e., without application of a timespan), the greatest interest concerns the area of "Information & ICT", followed by "Education". Part of the attention is devoted to "Buildings", whereas, at the end, there is "Culture", and "Collections" slightly ahead of it. More than a quarter of the works deal with the topics identified in the area of "Other". This last and heterogeneous group turned out to be surprisingly large, which may indicate that it deserves a separate analysis, which can constitute one of the possible future research directions. At this point, it can be tentatively said that the publications which were assigned to this group dealt mainly with the overall picture of a library activity, and practices that are aimed at improving the library's performance within the area of SDGs accomplishment. There was also a number of papers which focused on the most important aspects of SDGs implementation

in developing countries, in the context of, and in connection with, the possibilities created by libraries as information centers.

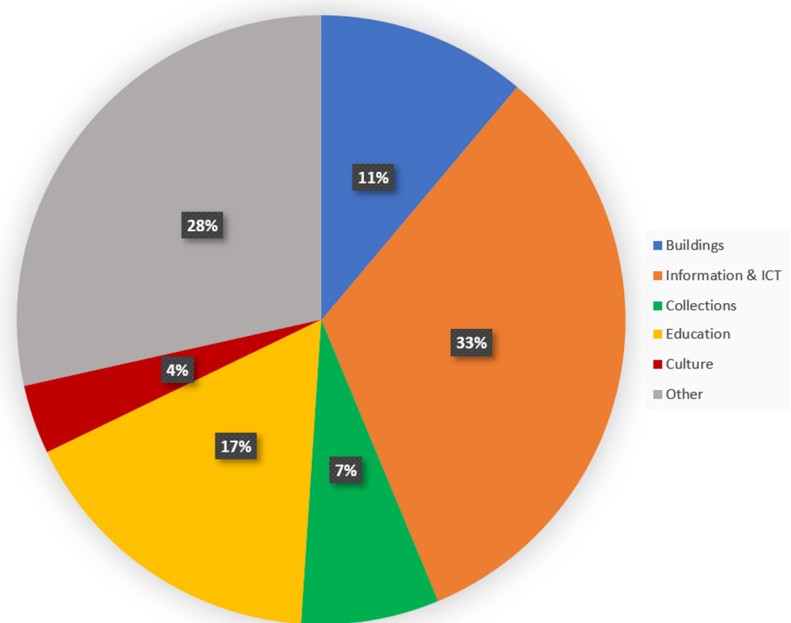

**Figure 2.** The proportions of scientific interest between individual topical areas.

A closer look from the time flow perspective is shown in Figure 3. As previously explained, the timeline ends here in 2020, as the data for 2021 is incomplete, which would bias the results. On the other hand, the timeline starts with 2000, since, before this date, there were very few scientific works (eight works in total, and the first one published in 1994). At first glance, it can be seen here how much information technologies, including.

ICT, have been of key importance from the very beginning of their development. The area of "Education" seems to be slightly correlated with the "Information & ICT" area, and is usually third right after the topical area of "Other". The area of "Culture" is relatively poorly represented by scientific works, and although initially it is of practically no interest, it is clearly increasing in importance after 2016. After 2019, there is a clear decline in interest in the area of "Buildings", and a further increase in interest in "Information & ICT". It can be cautiously assumed that this state of affairs is caused by the challenges brought by the global COVID-19 pandemic.

A slightly different perspective is provided by looking at the same data, but in relative terms. Figure 4 shows the percentage share of individual topical areas over time. This time, we are witnessing intermittent declines in the "Information & ICT" area. This means that although the frequency of taking up this topic increases year by year, its percentage share in relation to other topics is decreasing, but in the end, in 2020, it is the highest among all the others anyway. The year 2020 seems to bring certain polarization: behind "Information & ICT", the "Other" and "Education" areas are in a similar place, and 10% of the interest cannot exceed the areas of "Buildings", "Collections", and "Culture".

According to the authors, it may be interesting to look at the relations between individual topical areas in terms of their concurrence in scientific works. To carry out such research, previous assignments to topical areas were used, regardless of whether the given topic was considered as the main or side. The graph reflecting these relations is presented in Figure 5. Since the co-occurrence relation is a symmetric relation in the understanding of abstract algebra, this graph is an undirected one. However, since the edge thickness linking two nodes shows the strength of the relationship between nodes, it is a weighted graph. For each pair of topical areas, the number of scientific papers in which they are undertaken simultaneously was counted. In this way, the weights of the individual edges

were calculated. Moreover, the distribution of the vertices representing the individual topical areas is not accidental—an algorithm has been applied here that takes into account the strength of relationships between individual vertices resulting from the weights of the edges connecting them. Thus, the closer the given vertices on the graph are to each other, the more they are related. In addition, the size of the nodes is not accidental, and it results from the weighted degree, i.e., the more and the thicker the edges enter a given vertex, the larger its size. Considering the above, and looking at the largest nodes and the thickest edges, interesting conclusions can be drawn.

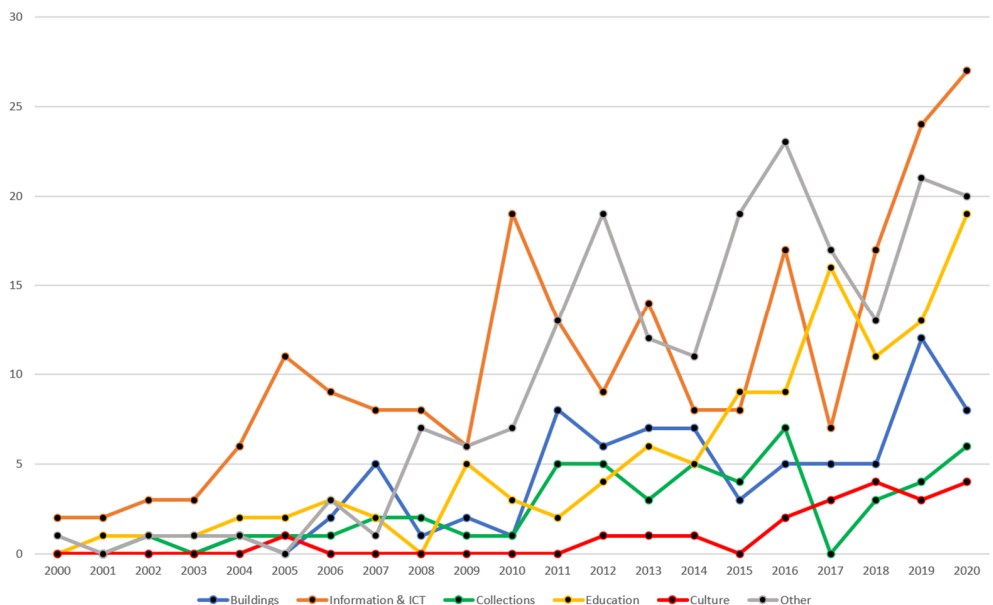

**Figure 3.** The volume of interest (number of scientific works) of topical areas over time.

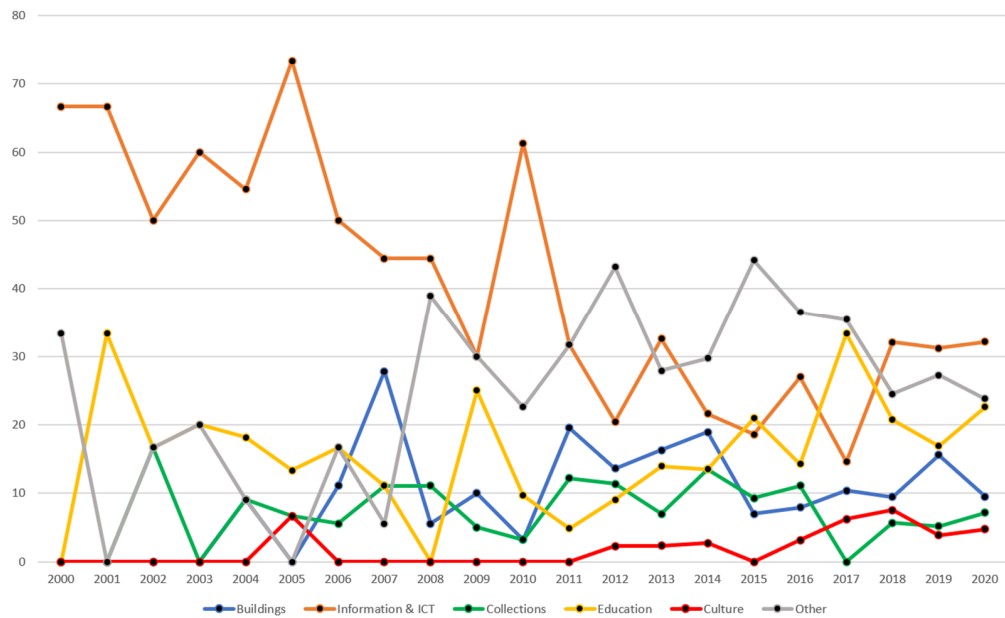

**Figure 4.** The relative volume (percentage share) of interest of topical areas over time.

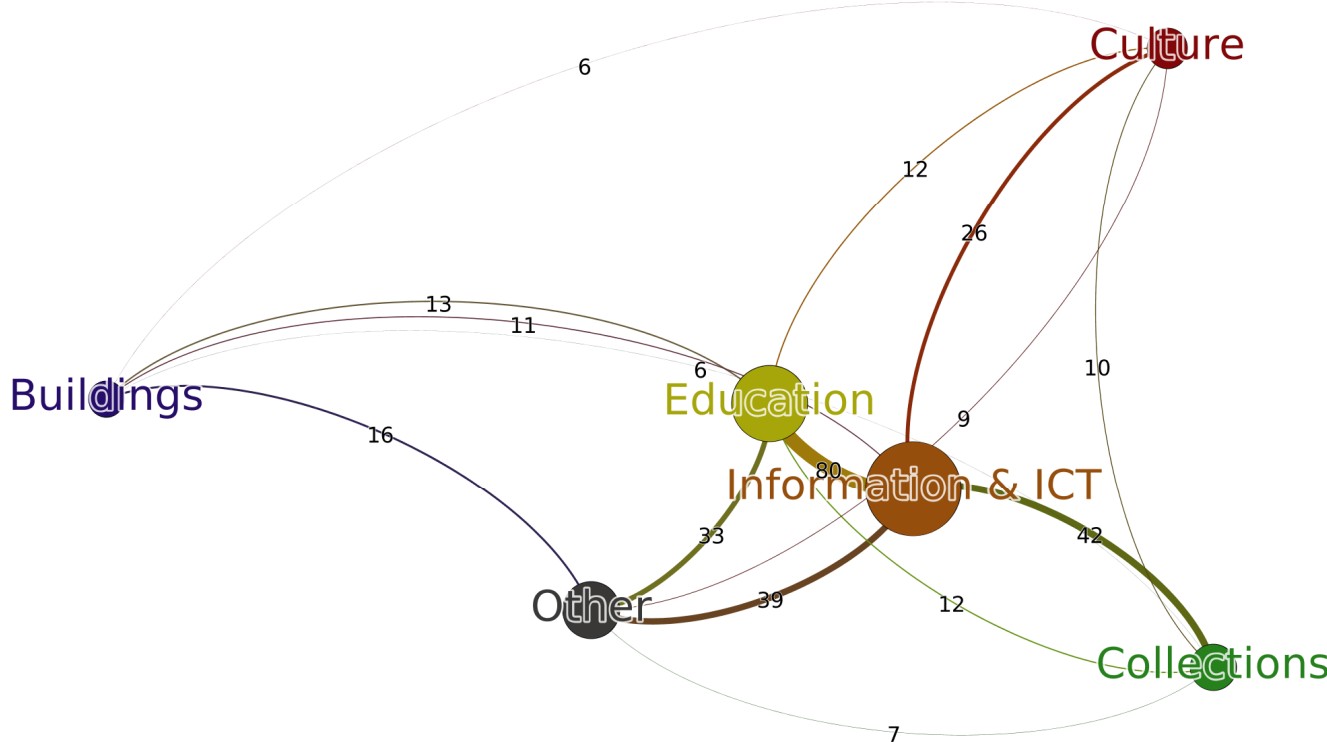

**Figure 5.** Relationships between topical areas in terms of co-occurrence.

At first glance, the figure in question shows that the most frequently simultaneously discussed topics are the areas of "Information & ICT" and "Education". The area of "Information & ICT" is also a winner regarding coexisting with other topics, and it is also discussed (apart from the aforementioned "Education") often with "Collections", "Culture", and "Other", whereas it is least related to "Buildings", which, incidentally, turns out to be the area that is the most hermetic. Additionally, taking into account the metric of the weighted degree of the nodes, it turns out that the area of "Culture" is ahead of the area of "Buildings".

Finally, as this is a common part of similar bibliographic research found in the literature, a keyword cloud was constructed, and is shown in Figure 6. Three approaches were used to build the keyword cloud model. Firstly, for the sake of clarity, the number of keywords presented has been limited to the first dozen in terms of frequency. Second, terms that are synonymous have been merged. Third, the size of a given keyword is directly proportional to its frequency after merging. All of that, according to the best intentions of the authors, should result in a simple, but self-explaining and contributing, visualization.

Despite the best efforts of the authors as to the objectivity and transparency of the conducted research, some doubts may arise, as there are no common standards or guidelines for conducting this type of research. There are two issues in regards to the limitation of this study. The first one is the timespan of the collected data. The search query was executed right after June 2021, so the year of 2021 has incomplete data. For this reason, the interpretation of the trend analyses does not take into account the year 2021. However, there is nothing to prevent the inclusion of data from the first half of 2021 in the global analysis (those not related to the division into periods). On the other hand, it is also worth mentioning that the data in the Scopus database is indexed continuously, and with a certain delay. Therefore, the query execution in mid-2021 is intended, and, thus, should result in high completeness of data from previous years, up to 2020. The second limitation is related to the search query definition, and, specifically, to how precisely it defines the dataset reflecting the subject matter, which was justified in Section 2 already.

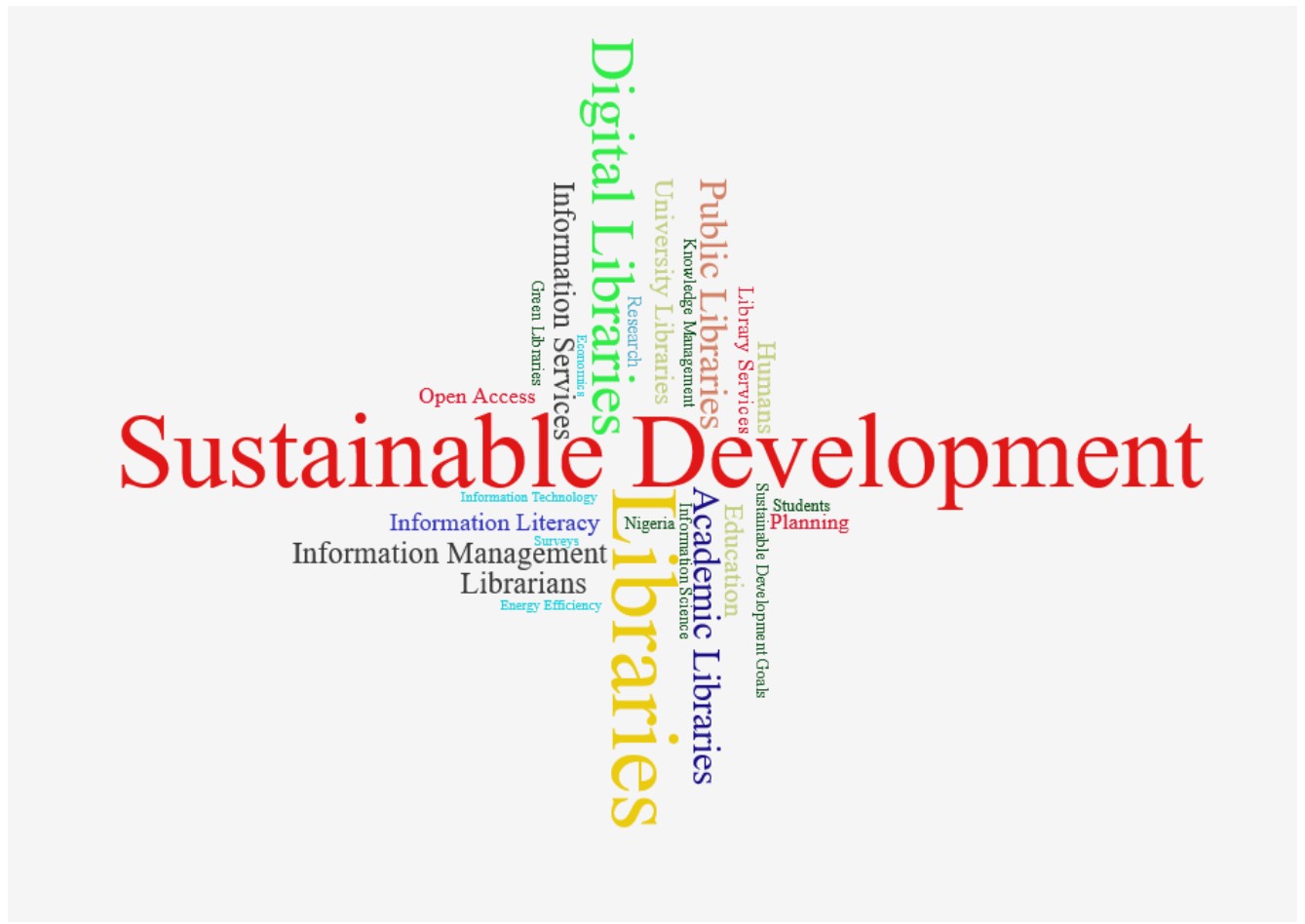

**Figure 6.** The word cloud of keywords.

Summing up, it is worth emphasizing the key findings of the research conducted as part of this paper:

- The number of scientific works in the field of sustainable development seen from the LIS perspective grows almost continuously year by year;
- The topical area of the greatest interest of the part of the scientific world which explores the sustainability issues of LIS, is currently the area of "Information & ICT"; despite the fact that it is the youngest area, it is the most intensively research;
- The most frequently co-occurring in the scientific literature of the sustainable LIS topical areas are "Information & ICT" together with "Education";
- The area of "Buildings" forms the most independent topic.

Generally speaking, the sustainability and sustainable development concept seems to interfuse each LIS topic that has been identified by the authors on the basis of their systematic literature review. It can supposedly be understood as a reflection of a growing concern about "our common future", including libraries as cultural, educational, and information centers. It is especially evident in relation to the social pillar of sustainability, mainly through libraries' contribution to the establishment of a participatory and sustainable information society. The economic pillar is, in turn, reverberated in a sustainable approach to maintaining, managing, storing, and developing library collections, and making library services more effective and efficient. The environmental pillar takes the form of sustainable library buildings' design, and taking care of GHG emissions generated during their exploitation, and standards of ICT usage. The last pillar, namely a cultural one, is associated with cultural heritage preservation, making it accessible for the public, and with some elements of architectural design or the redesign of library edifices. These pillars are

also interconnected in a variety of ways, which creates numerous opportunities for further research and discoveries of new methods of attaining SDGs with the aid of the whole LIS specialist community.

To shed further light on the landscape of sustainability on the subject matter, it would be worth analyzing the activity of individual countries, and the scientific collaboration patterns at the level of countries, institutions, and individual scientists.

Also, a more detailed breakdown of the "Other" category into subcategories was difficult to implement in the first run of topical tagging done by the authors so far, as without a quantitative picture of the individual candidate subcategories, it was difficult to include a given subcategory.

Therefore, the authors will take a closer look at the "Other" topical area once again, divide it into subcategories, and analyze it quantitatively.

These, together with the collaboration analysis, will be the directions of further research of the authors.

**Author Contributions:** Conceptualization, A.M.K., Ł.O. and Ł.W.; methodology, A.M.K., Ł.O. and Ł.W.; software, A.M.K. and Ł.W.; validation, A.M.K. and Ł.O.; writing–original draft preparation, A.M.K., Ł.O. and Ł.W.; writing–review and editing, A.M.K. and Ł.O.; visualization, A.M.K. and Ł.W.; funding acquisition, A.M.K. All authors have read and agreed to the published version of the manuscript.

**Funding:** This work was supported by University of Silesia in Katowice (Institute of Culture Studies).

**Institutional Review Board Statement:** Not applicable.

**Informed Consent Statement:** Not applicable.

**Data Availability Statement:** Publicly available datasets were analyzed in this study. This data can be retrieved here (using the search query provided in this paper): [https://www.scopus.com/](https://www.scopus.com/), accessed on 15 November 2021].

**Conflicts of Interest:** The authors declare no conflict of interest.

## Abbreviations

The following abbreviations are used in this manuscript:

| | |
|---|---|
| ASRS | Automated Storage and Retrieval System |
| DL | Digital Library |
| GHG | Greenhouse Gases |
| GLAM | Galleries, Libraries, Archives, and Museums Sector |
| IAP | IFLA International Advocacy Programme |
| ICT | Information and Communications Technology |
| IFLA | International Federation of Library Associations and Institutions |
| IL | Information Literacy |
| IR | Information Retrieval |
| IS | Information Systems |
| ISKO | International Society for Knowledge Organization |
| IT | Information Technology |
| KOS | Knowledge Organization Systems |
| LEED Leadership in Energy and Environmental Design Certificate LIS | Library and Information Science |
| PCSIS | Participatory, Co-operative, Sustainable Information Society |
| SDG | Sustainable Development Goals |
| SLR | Systematic Literature Review |
| ULSF | Association of University Leaders for a Sustainable Future |
| UN | United Nations |
| URL | Uniform Resource Locator |

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
