# Peer review of "The Landscapes of Sustainability in the Library and Information Science: Systematic Literature Review"

_sustainability, doi:10.3390/su14010441_

Round 1

Reviewer 1 Report

This is a very interesting and wide-ranging overview. The authors provide a detailed analysis and thorough recounting of the applications of the term ‘sustainability’ as it appears in the disciplinary literature. The introduction provides a well-researched, succinct description of the history of the sustainability movement for libraries and information sciences, with a lucid use of the existing literature that continues throughout the paper.

The search query used by the authors is intentionally imprecise. The authors provide an excellent justification for the selected query toward the end of the paper. It is suggested that the justification on pages 23-24 be moved to the methods section so that it is clear that the decision not to incorporate synonymous or related terms was intentional.

The authors have chosen a broad conceptual framework with which to understand sustainability. They connect concepts like resource sharing between libraries, green buildings, information literacy, and collections to sustainability very clearly. While they tie most of the literature they discuss to at least one of the pillars of sustainability, there are some discussions that might benefit from a clearer link to sustainability. As examples, the sections on big deals on page 11, the Hayman and Smith article on page 15, and the drawbacks of an LIS degree on page 16 all discuss the sustainability of practices in or characteristics of the discipline, but might be more closely connected to environmental, social, economic, or cultural sustainability.

Finally, the authors have classified a significant proportion of the articles as ‘other.’ I wonder if it’s possible to recategorize some of that section using a more meaningful description(s). The section on page 19-20 points to the role of libraries in democratization and social/research development. Is there a way to pull that out in a way that is more descriptive than ‘other?’ This would be particularly useful for the analysis in section 4, where the ‘other’ category leaves questions about how those articles connect to the other categories.

Reviewer 2 Report

The topic is interesting and worthy of presentation in a publication, but this manuscript is too long and needs to be more concentrated around higher education, academic libraries, and information studies.  Mixing the concepts of sustainable development with sustainability and conservation with preservation is confusing and needs to be clarified. Instead of using general terms such as human species, "other topical area" the authors should focus this manuscript on specific factors contributing to different perception of sustainability in various cultures.   

Reviewer 3 Report

This paper presents on an interesting topic and reports some compelling findings. Overall, the methods and reporting seem appropriate. I would, however, recommend a few revisions:

  • There are many very long and bulky paragraphs. Particularly on pages 7 through 20, there are paragraphs that appear to stretch for whole pages. These need to be broken down significantly to improve readability. You do not necessarily need to remove any content, but may make some minor revisions to have shorter paragraphs that still effectively communicate the main messages.
  • In the results section, the authors mention “the previous chapter.” However, my understanding is that this submission is supposed to be an article, not a chapter. I’m not sure what this reference means, then. I would recommend revising this section in order to remove these mentions.
  • I thought more about the topic of climate change and information services would be included, considering the title, abstract, and subject area. As such, I thought some of the following could have been included (note: I am an author on one of these papers, but I am not suggesting that the authors cite all of these, just pick a few that are relevant to their work to demonstrate their knowledge of existing scholarship in the area – or, alternatively, provide some justification as to why they are not relevant):
    • Prince, J. D. (2012). Climate change in libraries: library functions move to the cloud. Journal of electronic resources in medical libraries9(1), 87-93.
    • Charney, M., & Hauke, P. (2020). Global action on the urgency of climate change: Academic and research libraries’ contributions. College & Research Libraries News81(3), 114.
    • Murgatroyd, P., & Calvert, P. (2013). Information-seeking and information-sharing behavior in the climate change community of practice in the pacific. Science & Technology Libraries32(4), 379-401.
    • Lund, B. (2019). Barriers to ideal transfer of climate change information in developing nations. IFLA journal45(4), 334-343.
    • Siyao, P. O., & Sife, A. S. (2021). Sources of climate change information used by newspaper journalists in Tanzania. IFLA journal47(1), 5-19.
    • Miller, K. (2010). Public libraries going green. Chicago, IL: American Library Association.
  • On line 1064, why have you abbreviated figure to fig.? This seems unnecessary.
